# Deep Learning for Model Correction in Cardiac Electrophysiological Imaging

**Victoriya Kashtanova**[1,2]                                   VICTORIYA.KASHTANOVA@INRIA.FR
[1] *Inria, Université Côte d'Azur, Nice, France*
[2] *3IA Côte d'Azur, Sophia Antipolis, France*

**Ibrahim Ayed**[3,4]
[3] *Sorbonne University, Paris, France*
[4] *Theresis lab, Thales, France*

**Andony Arrieula**[6,7,8]
[6] *CARMEN Research Team, Inria Bordeaux – Sud-Ouest, Talence, France*
[7] *IHU Liryc, Fondation Bordeaux Université, Pessac, France*
[8] *Université de Bordeaux, IMB, UMR 5251, Talence, France*

**Mark Potse**[8,7,6]

**Patrick Gallinari**[3,5]
[5] *Criteo AI Lab, Paris, France*

**Maxime Sermesant**[1,2]                                     MAXIME.SERMESANT@INRIA.FR

**Editors:** Under Review for MIDL 2022

## Abstract

Imaging the electrical activity of the heart can be achieved with invasive catheterisation. However, the resulting data are sparse and noisy. Mathematical modelling of cardiac electrophysiology can help the analysis but solving the associated mathematical systems can become unfeasible. It is often computationally demanding, for instance when solving for different patient conditions. We present a new framework to model the dynamics of cardiac electrophysiology at lower cost. It is based on the integration of a low-fidelity physical model and a learning component implemented here via neural networks. The latter acts as a complement to the physical part, and handles all quantities and dynamics that the simplified physical model neglects. We demonstrate that this framework allows us to reproduce the complex dynamics of the transmembrane potential and to correctly identify the relevant physical parameters, even when only partial measurements are available. This combined model-based and data-driven approach could improve cardiac electrophysiological imaging and provide predictive tools.

**Keywords:** Physics-based learning, Electrophysiology, Deep learning, Simulations

## 1. Introduction

Mathematical modelling of the heart has been an active research area for decades, and it is now more and more coupled with artificial intelligence approaches (Mansi et al., 2020; Giffard-Roisin et al., 2017; Karoui et al., 2021; Trayanova et al., 2021). Among the multi-physics phenomena involved in cardiac function, cardiac electrophysiology (EP) models can accurately reproduce electrical behaviour of cardiac cells.

In order to describe the dynamics of transmembrane voltage, current, and different ionic concentrations in the cardiac cell, biophysically detailed models such as the Ten Tusscher-Panfilov model (Ten Tusscher et al., 2004; Ten Tusscher and Panfilov, 2006) have been proposed. However, these models are complex and computationally expensive, and have many hidden variables which are impossible to measure, making model parameters difficult to personalise. Another type of model, phenomenological models, are simplified models derived from biophysical models. Examples include the FitzHugh-Nagumo, Aliev-Panfilov, and Mitchell-Schaeffer models (FitzHugh, 1961; Nagumo et al., 1962; Aliev and Panfilov, 1996; Nash and Panfilov, 2004; Mitchell and Schaeffer, 2003). These models employ fewer variables, they have fewer parameters and are therefore especially useful for rapid computational modelling of wave propagation at the organ level. However, they are less realistic and therefore need a complementary mechanism to fit them to the measured data. Machine learning and in particular deep learning (DL) approaches could help providing such a correction mechanism. The combination of rapid phenomenological models and machine learning components could then allow the development of rapid and accurate models of transmembrane dynamics.

In the last few years, DL neural networks have been increasingly used in order to learn dynamical models from data motivating a large number of publications. For example, Long et al. (2018, 2019) endowed neural layers with additional structure, useful for learning PDEs. Chen et al. (2018) used the adjoint method to learn differential equations parametrised with neural networks. Ayed et al. (2019b) propose a framework for learning models using a purely data driven approach in partially observable settings. Willard et al. (2020) propose a broad survey of ML for physics-based modeling.

In spite of achieving good progress in cardiac electrophysiology simulations (Ayed et al., 2019a; Kashtanova et al., 2021) data-driven models alone could not reproduce the complex unseen dynamics like the repolarisation (Kashtanova et al., 2021), and the maximum forecasting horizon is still limited. For this reason, researchers have begun to use coupled physico-statistical approaches for cardiac electrophysiology simulations with a high precision and at low cost. For example, Court and Kunisch (2021) designed a neural network that approximates the FitzHugh-Nagumo model, Sahli Costabal et al. (2020) used a physics-informed neural network for cardiac activation mapping accounting for underlying wave propagation dynamics, Fresca et al. (2021) proposed an approach to create a nonlinear reduced order model with the help of deep learning algorithms (DL-ROM) for cardiac electrophysiology simulations, and Herrero Martin et al. (2022) present a physics-informed neural network for accurate action potential simulation and EP model parameter estimation. However, the majority of these coupled approaches bases on high-fidelity physical models and fits them to the data. This could be computationally expensive and cannot manage large discrepancies between simulated and real data.

To alleviate this limitation, we propose a framework to Augment incomplete PHYsical models with a deep learning component for ideNtifying comlex cardiac ElectroPhysiology dynamics (APHYN-EP) from data, based on a fast low-fidelity (or incomplete) physical model. This framework has two components which decompose the dynamics into a physical and a data-driven term. The data-driven deep learning component is designed so as to capture only the information that cannot be modeled by the incomplete physical model. The proposed model closely follows the approach of Yin et al. (2021). But in contrast

to this work, that considers fully-observable dynamics and simple test use cases, cardiac electrophysiology dynamics have a high complexity and represent simultaneously multiple underlying processes. Furthermore, most cardiac electrophysiology models lack measurements for some variables, which makes them partially-observable and requires inferring the dynamics from incomplete observations only. Fig. 1 presents the general framework of our approach. Training amounts to identifying the physical model parameter (inverse problem) and learning the neural network parameters (direct problem) together. After training, the model can be used for forecasting at multiple horizons.

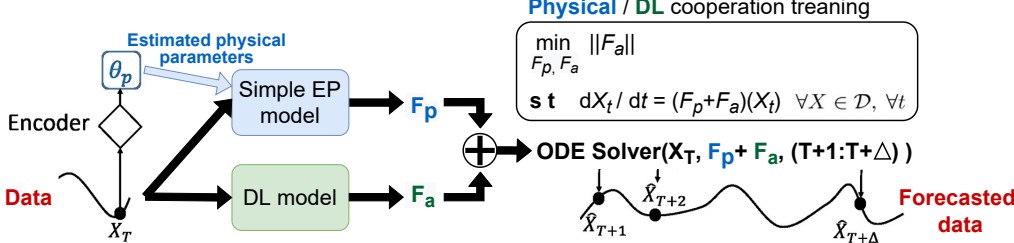

Figure 1: General APHYN-EP framework scheme. During the training phase two-component framework learn the parameters for the physical ($F_p$) and the data-driven ($F_a$) components from data. Then via an ODE solver the framework can forecast further the learned dynamics.

## 2. Learning Framework

To learn cardiac electrophysiology dynamics ($X_t$) we solve an optimization problem via a physics-based data-driven framework APHYN-EP. This framework combines a physical model ($F_p$) representing an incomplete description of the underlying phenomenon and a neural network ($F_a$) which will complement the physical model by capturing the information that cannot be modeled by the physics component:

$$\min_{F_p \in \mathcal{F}_p, F_a \in \mathcal{F}_a} \|F_a\| \text{ subject to } \forall X \in \mathcal{D}, \forall t, \frac{dX_t}{dt} = F(X_t) = (F_p + F_a)(X_t). \tag{1}$$

Our incomplete physical model is the two-variable ($v, h$) model by Mitchell and Schaeffer (2003) for cardiac electrophysiology simulation (2). The variable $v$ represents normalised ($v \in [0, 1]$) dimensionless transmembrane potential while the "gating" variable $h$ controls the repolarisation (return to the initial state):

$$\partial_t v = \text{div}\left(\sigma \mathbf{I} \nabla v\right) + \frac{h v^2 (1 - v)}{\tau_{\text{in}}} - \frac{v}{\tau_{\text{out}}} + J_{\text{stim}}$$

$$\partial_t h = \begin{cases} \frac{1-h}{\tau_{\text{open}}} & \text{if } v < v_{\text{gate}} \\ \frac{-h}{\tau_{\text{close}}} & \text{if } v > v_{\text{gate}} \end{cases} \tag{2}$$

where $J_{\text{stim}}$ is a transmembrane potential activation function, which is equal to 1 in the stimulated area during stimulation time ($t_{\text{stim}}$). This physical model has been successfully

used in patient-specific modelling (Relan et al., 2011), it covers general electrophysiology dynamics and, in contrast to more detailed electrophysiology models, it is flexible in terms of spatial and temporal steps. Assuming that we can obtain the coordinates of an applied electrical stimulation from the data and using $v(t = 0) \equiv 0$ and $h(t = 0) \equiv 1$ we can calculate an approximation of $h$ for any time point $t$ with the help of a simple integration scheme.

For the data-driven component we use a Residual Network (ResNet) (He et al., 2016), because it can accurately reproduce transmembrane potential dynamics (Ayed et al., 2019a; Kashtanova et al., 2021). The choice of data-driven component is not limited by the ResNet architecture. We performed preliminary experiments with other types of convolutional networks (out of scope for this paper), but overall the ResNet model was the more stable along the different simulations.

Instead of solving the ODE in Eq. (1), we use an integral trajectory-based approach which is robust and less sensitive to the time resolution (Yin et al., 2021). We compute the next state $\tilde{X}_{h\Delta t}^{(i)}$ from the initial state $X_0^{(i)}$ as an approximate solution of the integral $\int_{X_0^{(i)}}^{X_0^{(i)}+h\Delta t}(F_p^{\theta_p} + F_a^{\theta_a})(X_s)\,dX_s$ obtained by a differentiable ODE solver (Chen et al., 2018, 2021). The APHYN-EP training uses an algorithm adapted from Yin et al. (2021).

---

**Algorithm 1:** APHYN-EP

---

Initialization: $\theta_0, \lambda_0 \geq 0, \tau > 0$;
**for** $epoch = 1 : N_{\text{epochs}}$ **do**
    **for** $batch\ in\ 1 : B$ **do**
        $\mathcal{L}_{\text{traj}}(\theta_j) = \sum_{i=1}^{N} \sum_{h=1}^{T/\Delta t} ||X_{h\Delta t}^{(i)} - \tilde{X}_{h\Delta t}^{(i)}||$
        $\theta_{j+1} = \theta_j - \nabla\left[\lambda_j \mathcal{L}_{\text{traj}}(\theta_j) + ||F_a||\right]$
    **end**
    $\lambda_{j+1} = \lambda_j + \tau \mathcal{L}_{\text{traj}}(\theta_{j+1})$
**end**

---

Additionally, in order to train simultaneously the physical and the data-driven components of APHYN-EP via automatic differentiation tools (provided by the Pytorch library (Paszke et al., 2019)) we implemented the Laplace operator in (2) with a simple finite-difference scheme. To avoid difficulties with high time resolution required in this numerical scheme we used two different time steps in the integration schemes for the physical component computations and for the computations of the final forecast given by the framework.

## 3. Experimental settings

### 3.1. Data collection

To evaluate our method, we used a dataset of transmembrane potential activation simulated with a monodomain reaction-diffusion equation and the Ten Tusscher – Noble – Noble – Panfilov ionic model (Ten Tusscher et al., 2004), which represents 12 different transmembrane ionic currents. The simulations were performed with a recent version of the propag-5 software (Krause et al., 2012; Potse, 2018) with a spatial step of 0.2 mm and a time step of

1 ms, the same as used by Ten Tusscher et al. (2004). The computational domain represents a slab of 2D cardiac tissue of size $24 \times 24$ elements. For one data sample, one stimulation was applied for 1 ms in the selected area for transmembrane potential activation. Since the grid was symmetric under 90-degree rotations, the stimulations were done only on each grid point of the first quarter of the cardiac slab. Then, we applied data-augmentation techniques to translate the simulations on the three remaining quarters. Overall we had a database of 500 training samples and 100 validation samples. Each simulation represented 500 ms of propagation, which represents 40 seconds of computation on a 12-core Intel Xeon W-2133 CPU.

The data simulated via the Ten Tusscher model are considered here as the ground truth. The objective is then to learn the complex dynamics generated with this model with the APHYN-EP model combining a simplified physics description with a deep learning component. This will result in a low computational cost surrogate model of the computationally intensive Ten Tusscher model.

### 3.2. Training settings

The physical model ($F_p$) of Eq. 2 was implemented with a standard finite-difference scheme for the Laplace operator with a spatial resolution of 1 mm$^2$ pixels and inner time resolution of 0.01 ms. We estimated only $\sigma$ and $\tau_{\text{in}}$ as unknown parameters in (2), because they represent the major part of early dynamics (velocity and upstroke) and therefore the main difference between the Mitchell–Schaeffer and the Ten Tusscher – Noble – Noble – Panfilov models in our simulations. The initial Mitchell-Schaeffer model parameters are taken as in the original paper (Mitchell and Schaeffer, 2003): $\tau_{\text{out}} = 6$, $\tau_{\text{open}} = 120$, $\tau_{\text{close}} = 150$, $v_{\text{gate}} = 0.13$ and $t_{\text{stim}} = 1$. The data-driven model ($F_a$) is ResNet with 8 filters at the initial stage, one down-sampling initial layers and three intermediary blocks, and starts with a re-weighted orthogonal initialisation for its parameters.

We used a time resolution of 0.1 ms to compute the forecast given by APHYN-EP. Training was performed using a horizon of 6 time frames. Training was performed by learning a 4 frames horizon for the first 10 epochs, and then 6 frames. This leads to more stable results. We trained APHYN-EP until full model convergence (about 100 epochs) using an ADAM optimiser (Kingma and Ba, 2014) with initial learning rate of $10^{-3}$. The algorithm hyper-parameters $\lambda_0$ and $\tau$ were set to 1 and $10^3$ respectively.

The code and data used for APHYN-EP training are freely available on the official github project page.

## 4. Results

We present here qualitative results on the forecast over 9 ms after assimilating only one first frame of dynamics (see Fig. 2). These first 9 ms represent an important part of the cardiac dynamics from early depolarisation to full depolarisation. We can observe a very good agreement between the ground truth and the forecast transmembrane potentials generated by APHYN-EP in Fig. 3. The correction term brought by $F_a$ is clearly visible.

Table 1 shows the mean squared error (MSE) results for our framework for different forecasting horizons on validation data samples. To calculate this error, for each data sample, we fed the model with only one initial test measurement, then let it predict several

Table 1: Mean-squared error (MSE) of normalised transmembrane potential forecasting per time-step for different forecasting horizons.

|  | MSE (6 ms) | MSE (12 ms) | MSE (24 ms) | MSE (50 ms) |
|---|---|---|---|---|
| APHYN-EP | 0.0057 | 0.0037 | 0.0029 | 0.002 |
| Physical model only | 0.0093 | 0.0111 | 0.0096 | 0.0085 |
| Resnet model only | 0.0195 | 0.0220 | 0.1593 | 9.9212 |

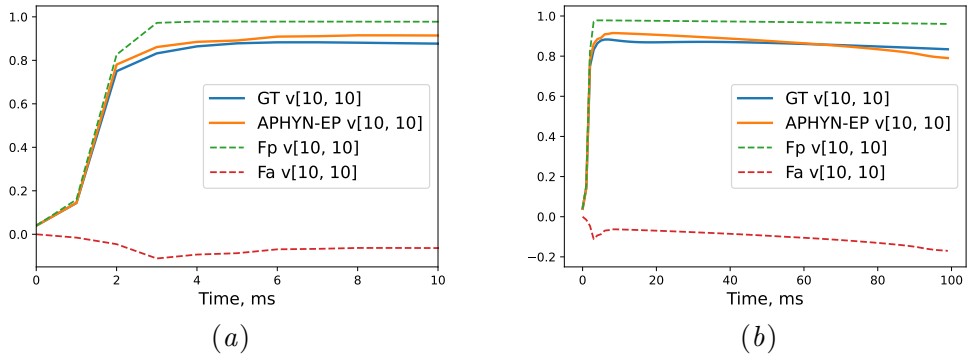

Figure 2: APHYN-EP predicted dynamics for the transmembrane potential diffusion. The figure shows a 9 ms of forecast).

Figure 3: (a,b) Transmembrane potential at point (10,10) in the cardiac slab (red point, see Fig. 2) with different forecasting horizons. Ground truth (GT), APHYN-EP, physical ($F_p$) and data-driven ($F_a$) component of APHYN-EP.

steps forward without any additional information. We also added for comparison two baseline models corresponding to the two components of our model, each used alone: only the "incomplete" physical model and only the data-driven model (ResNet) trained on the same dataset as APHYN-EP, described in 3.1. As we can see, APHYN-EP captured the observed

dynamics with good precision for different time horizons, even if for training we used only the first 6 ms. In the same time, the pure physical and the data-driven models when used alone struggle to learn the proper dynamics. Figure 5 visually confirms those numerical results. The physical model (as well as APHYN-EP) correctly identifies the velocity and the activation time of transmembrane potential, but not the transmembrane potential values, due to its physical construction and limitations. The data-driven model can have a good precision, but it reproduces only average dynamics and is very sensitive to self-generated noise, which is crucial when forecasting.

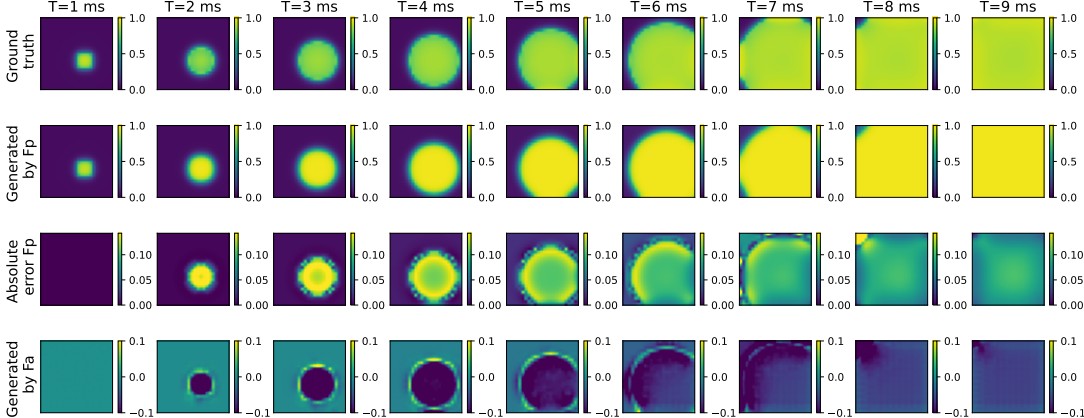

Figure 4: Predicted dynamics for the transmembrane potential diffusion by APHYN-EP physical component (second row), error with ground-truth diffusion for this physical component of APHYN-EP (third row), and trained APHYN-EP data-driven component contribution (bottom row).

Figures 3 and 4 represent the performance of different components of APHYN-EP and their contribution to the final result. We can see which part of the generated transmembrane potential was created by the physical component of the framework (see Fig. 4 (second row)). The data-driven component was used only to correct the difference between the ground-truth dynamics and the physical part (see Fig. 4 (third and fourth rows)).

## 5. Conclusion

We have presented the APHYN-EP framework for modeling complex cardiac electrophysiology dynamics via a surrogate model combining simplified physics and deep neural networks. We demonstrated that this framework is able to reproduce with good precision the dynamics simulated by the Ten Tusscher – Noble – Noble – Panfilov ionic model, even using a simplified electrophysiology model as a physical component of the framework.

Such framework opens up possibilities in order to introduce prior knowledge in deep learning approaches through explicit equations and to correct model errors from data.

Future work will evaluate this framework on more challenging settings: presence of scars, multiple onsets and various conduction velocities in the cardiac tissue slab. We also left for the future the adaptation of the APHYN-EP application on the surface of real 3D heart.

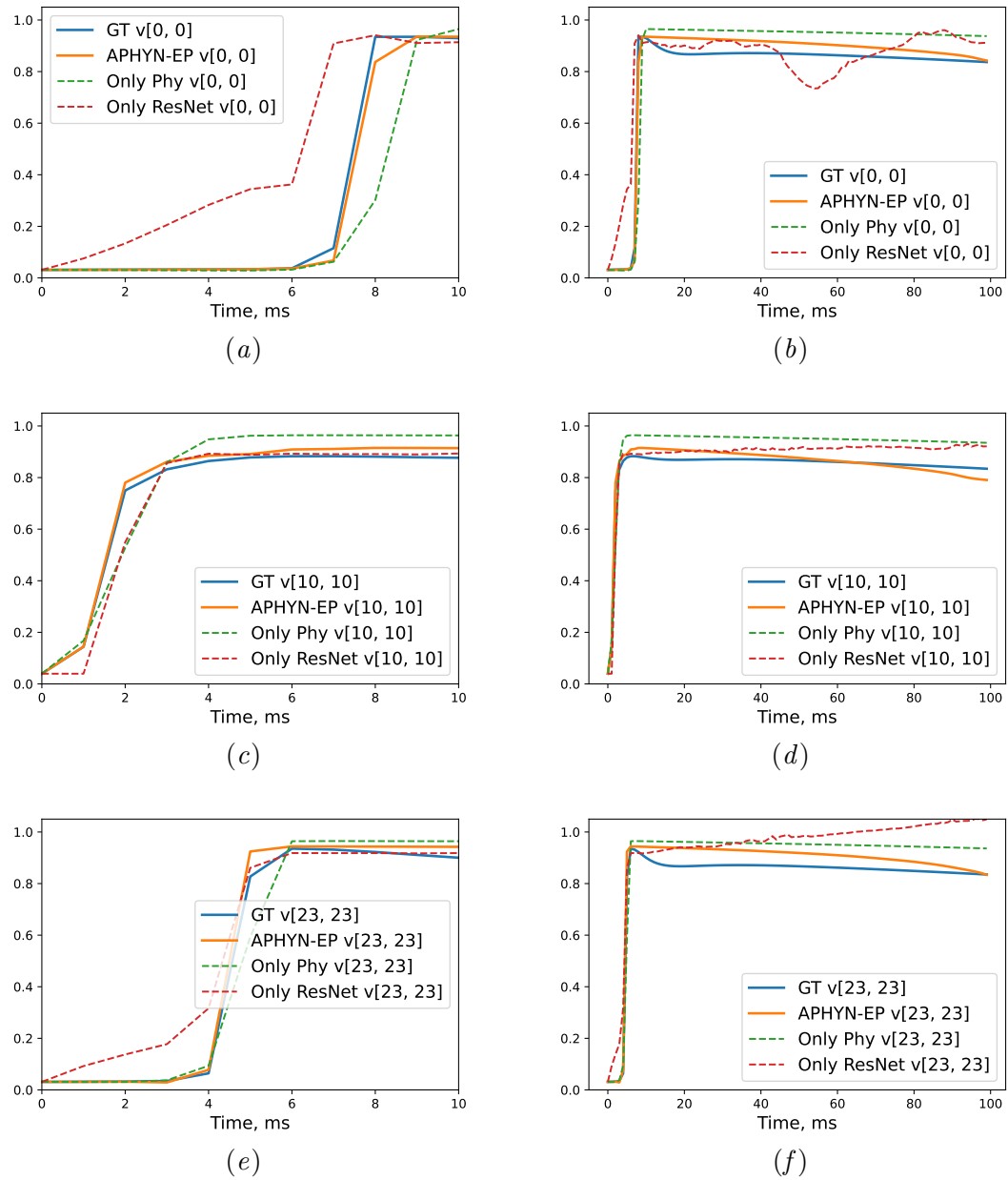

Figure 5: Transmembrane potential ground truth (GT), generated by APHYN-EP, by Physical model and ResNet model at the leftmost upper point (1,1) (a,b), at point (10,10) (c,d) and the rightmost bottom point (23,23) (e,f) in the cardiac slab with different forecasting horizons, the same GT dynamics as at Fig. 2-3.

## Acknowledgments

This work has been supported by the French government, through the 3IA Côte d'Azur Investments in the Future project managed by the National Research Agency (ANR) with

the reference number ANR-19-P3IA-0002, through the "Research and Teaching chairs in artificial intelligence (AI Chairs)" funding for DL4Clim project, and through ANR grant reference ANR-10-IAHU04-LIRYC. The research leading to these results has also received European funding from the ERC starting grant ECSTATIC (715093). The authors are grateful to the OPAL infrastructure from Université Côte d'Azur and to HPC resources of CINES under GENCI allocation 2020-A0090307379 for providing resources and support.

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
