# OpenReview forum: "Deep Learning for Model Correction in Cardiac Electrophysiological Imaging"
_MIDL.io/2022/Conference — MIDL 2022_

### Official Review · Reviewer_FsqR · 2022-01-12

**Confidence:** 3
**Preliminary Rating:** 4
**Recommendation:** Poster

**Summary:**

The authors propose a framework to solve cardiac electrophysiology problems that combine a (simple) physics solver with a deep learning  network to account for the effects not captured by the simple solver. The framework is tested on simulated cases, where it shows good agreement with ground truth. The results are promising but tested only in a simple simulation.

**Strengths:**

* The idea of combining data driven and physics driven terms, in such a complex multi-physics problem, is novel and very interesting for the community

* the paper is very well written and easy to follow, with a complete literature review, and a clear description of the methodology.

**Weaknesses:**

* The main weakness is the data employed, which is a rather simple simulation in a healthy setting. This is probably enough to prove the concept, but authors should explain more clearly how different these simulations are from real data, wht differences can be expected when switching to real images, and what is required to actually do that.

* Following with the point above, the paper is entitled "Deep Learning for Model Correction in Cardiac Electrophysiological Imaging" however since all data is simulated, there is really no "imaging" involved (except for the simulated patch which is a 2D array with constant value, hardly imaging data). Perhaps authors should indicate the simulation nature of the paper in the title to set fair expectations to readers.

* Authors claim that "According to restrictions posed on the data-driven (or deep learning) component, it only models information that cannot be captured by the physical model." However it is not clear or explicit in the paper what these restrictions actually are; it seems that these restrictions are only implicit by what the physical model employed can represent, so it would be a restriction on the physical model rather than on the data-driven component right?

**Deanonymize Review:**

yes

**Detailed Comments:**

* p3: Fig 1 is somewhat unclear, with a very succint caption. The caption should at least explain the meaning of the symbols, blocks and give an overall brief description. The text at the top of page 3 is also insufficient to clearly understand the figure.

* p3: "Where J_stim is a transmembrane potential activation function" this should start in lower case and not in a new paragraph since it continues from the eequation.

* p6: The table shows the Relative mean-squared error (MSE). Does relative mean it is in %?If so, numbers seem very close to 0, hence a statistical significance test is requires. It is relative to the ground truth, i.e. (val-gt)/abs(gt)^2? . The "relative" wording is indicated in the title but not in the text. If not relative, please indicate the units.

**Final Rating After The Rebuttal:**

4: Weak Accept

**Justification Of The Final Rating:**

Authors have addressed the comments partly: of my three main comments, I have not certainty they have addressed FIg 1. They have satisfactorily addressed the motivation part. And I don't think the issues with the data driven approach and the imaging component have really been addressed. So my original review holds mostly, and I still recommend a weak accept.

I think the paper is interesting, but the expectations raised from the title do not hold through the paper.

**Paper Type:**

methodological development

**Questions To Address In The Rebuttal:**

The main three things to address are:
1. to improve Fig 1 and its caption
2. Incorporate a more complete discussion on the simulation nature of the paper, how it relates to real cases and what are the limitations.
3. Clarify about the restrictions in the data-driven component as indicated in the Weaknesses section

**Special Issue:**

no

---

### Official Review · Reviewer_fWqy · 2022-01-22

**Confidence:** 3
**Preliminary Rating:** 3
**Recommendation:** Poster

**Summary:**

Solving the mathematical systems for cardiac electrophysiology modelling is often computationally demanding. This paper addresses the problem by proposing a new framework to model the dynamics of cardiac electrophysiology at lower cost.  A low-fidelity physical model is integrated with a learning component implemented via neural networks, which deals with quantities and dynamics that are neglected by the simplified physical model.


**Strengths:**

- The physical model is well explained. In particular, Fig. 1 is informative and provides a good overview of the proposed method.
- The idea of using neural networks to complement the physical model is interesting.

**Weaknesses:**

- The significance and motivation of the proposed method need to be better explained.
- The design of the data-driven model is somewhat arbitrary.
- There is a lack of comparison with existing methods.

**Deanonymize Review:**

no

**Detailed Comments:**

- Why is it important to reproduce the electrical behaviour of cardiac cells? The significance of the propose work can be better emphasized.

- In the introduction, existing neural network methods are described. The authors can clarify the limitation of these existing methods and how these limitations motivate the proposed method.

- How is the structure of the data-driven model determined? Is it optimal? What other options have the authors explored?

- The experiment seems simplistic. From Figs. 2 and 5, it seems that the ground truth is simply a growing circle. Does it reflect realistic tissue properties?

- In the experiments, the authors have not compared the results of the proposed method with those of existing coventional or neural network methods. It is not clear whether the proposed method is better than existing methods.

**Final Rating After The Rebuttal:**

3: Borderline

**Justification Of The Final Rating:**

Even for ResNet, there can be different versions with different widths and lengths. I am still not convinced that the current network structure in the proposed method is optimal, and this should be investigated extensively.

**Paper Type:**

methodological development

**Questions To Address In The Rebuttal:**

- The significance and motivation of the proposed method need to be better explained.
- A principled determination of the data-driven model is desired.
- The comparison with existing methods should be added.

**Special Issue:**

no

---

### Meta-Review · Area_Chair_LM5A · 2022-02-18

**Recommendation:** Accept (Poster)
**Confidence:** 4

**Metareview:**

This work is clear, very original, however it lacks a bit of significance and is more of a proof of concept. Here are the pros and cons I've identified.

Pros :
- This work deals with physics-based deep learning which is a very recent subject and one of growing interest for the medical community.
- Results on simulated data look promising.
- The theoretical part and the implementation is clear, too bad the code is not shared.

Cons :
- The responses to the reviewers are very dense, often repeat the same explanations from one reviewer to another and lack relevance. It is not clear what has been changed in the manuscript in particular.
- The proof of concept should have been to support in the claims of the authors, as well as the use of simulated data in the title as suggested by the last reviewer
- In order to make the work more robust and quantitative, different physical models could have been considered, different Resnet architectures and an application to real data.

This work is extremely interesting but still a little immature. So I suggest an acceptation as a poster.

---

### Decision · Program_Chairs · 2022-02-28

Accept